# Coordinating the morphogenesis-differentiation balance by tweaking the cytokinin-gibberellin equilibrium

Alon Israeli[1], Yogev Burko[1¤a], Sharona Shleizer-Burko[1¤b], Iris Daphne Zelnik[1¤c], Noa Sela[2], Mohammad R. Hajirezaei[3], Alisdair R. Fernie[4], Takayuki Tohge[4¤d], Naomi Ori[1]*, Maya Bar[1,2]*

1 The Robert H. Smith Institute of Plant Sciences and Genetics in Agriculture, Hebrew University, Rehovot, Israel, 2 Department of Plant Pathology and Weed Research, Plant Protection Institute, Agricultural Research Organization, Volcani Institute, Rishon LeZion, Israel, 3 Molecular Plant Nutrition, Department of Physiology and Cell Biology, Leibniz Institute of Plant Genetics and Crop Plant Research, Seeland, Germany, 4 Max-Planck-Institute of Molecular Plant Physiology, Potsdam-Golm, Germany

¤a Current address: Plant Biology Laboratory, Salk Institute for Biological Studies, La Jolla, California, United States of America
¤b Current address: Department of Medicine, University of California San Diego, La Jolla, California, United States of America
¤c Current address: Department of Biomolecular Sciences, Weizmann Institute of Science, Rehovot, Israel
¤d Current address: Graduate School of Biological Science, Nara Institute of Science and Technology (NAIST), Ikoma, Japan
* ori@mail.huji.ac.il (NO); mayabar@volcani.agri.gov.il (MB)

**Data Availability Statement:** All relevant data are within the manuscript and its Supporting Information files.

## Abstract

Morphogenesis and differentiation are important stages in organ development and shape determination. However, how they are balanced and tuned during development is not fully understood. In the compound leaved tomato, an extended morphogenesis phase allows for the initiation of leaflets, resulting in the compound form. Maintaining a prolonged morphogenetic phase in early stages of compound-leaf development in tomato is dependent on delayed activity of several factors that promote differentiation, including the CIN-TCP transcription factor (TF) LA, the MYB TF CLAU and the plant hormone Gibberellin (GA), as well as on the morphogenesis-promoting activity of the plant hormone cytokinin (CK). Here, we investigated the genetic regulation of the morphogenesis-differentiation balance by studying the relationship between LA, CLAU, TKN2, CK and GA. Our genetic and molecular examination suggest that *LA* is expressed earlier and more broadly than *CLAU* and determines the developmental context of CLAU activity. Genetic interaction analysis indicates that LA and CLAU likely promote differentiation in parallel genetic pathways. These pathways converge downstream on tuning the balance between CK and GA. Comprehensive transcriptomic analyses support the genetic data and provide insights into the broader molecular basis of differentiation and morphogenesis processes in plants.

**Funding:** The authors acknowledge funding from the Israel Science Foundation (2407/18 and 248/19 to NO), the Israel Binational Agricultural Research and Development fund (IS5103-18R to NO), and the Israel Binational Science Foundation (2015093 to NO). AI is grateful to the Azrieli foundation for the award of an Azrieli Fellowship. The funders had no role in study design, data collection and analysis, decision to publish, or preparation of the manuscript.

**Competing interests:** The authors have declared that no competing interests exist.

## Author summary

Morphogenesis and differentiation are crucial steps in the formation and shaping of organs in both plants and animals. A wide array of transcription factors and hormones were shown to act together to support morphogenesis or promote differentiation. However, a comprehensive molecular and genetic understating of how morphogenesis and differentiation are coordinated during development is still missing. We addressed these questions in the context of the development of the tomato compound leaf, for which many regulators have been described. Investigating the coordination among these different actors, we show that several discrete genetic pathways promote differentiation. Downstream of these separate pathways, two important plant hormones, cytokinin and gibberellin, act antagonistically to tweak the morphogenesis-differentiation balance.

## Introduction

Morphogenesis, originating from the Greek words *morphe*/shape and *genesis*/formation, is a fascinating biological process that has attracted human eyes since ancient times [1,2]. Several model systems have been used to study morphogenesis, from the first examination of chicken embryos by Aristotle [3–6]. Plants provide an excellent model system to investigate the shaping of an organism during the adult life cycle [7,8]. Despite the ancient origin of morphogenesis studies in both the animal and plant kingdoms, our understanding of the molecular mechanisms governing morphogenesis, in particular the connection between gene regulatory networks, function, and shape formation—is still not complete.

Aristotle's philosophy shaped our thinking of the term 'form' as fulfilling the full potential and destiny of oneself [3]. Leaves are vital photosynthetic, lateral organs produced by the plant throughout its life cycle. The development of plant leaves follows a common basic program, adjusted flexibly according to species, developmental stage and environment [9–12]. Morphogenesis and differentiation are important stages in leaf development, and the spatial and temporal balance between these processes influences leaf size and shape [10,13,14]. In compound leaved plants such as tomato, the ratio between these two stages favors longer morphogenesis, allowing for initiation of leaflets, resulting in the compound form [15]. The length of the morphogenetic window is thus a key determinant of final leaf shape. The flexibility of the morphogenetic window is regulated through a coordinated interplay between transcription factors and hormones [16–25]. Tomato leaf development is therefore an attractive system to investigate the contribution of the morphogenesis-differentiation balance to organ shaping.

The hormone gibberellin (GA) promotes leaf differentiation, while cytokinin (CK) promotes morphogenesis. Therefore the balance between the activity of these two hormones is key to the modulation of the morphogenesis-differentiation balance [17,26,27].

CIN-TCP transcription factors affect leaf shape by promoting differentiation, and maintenance of the morphogenetic window is dependent on low CIN-TCP activity during the early stages of leaf development [28–37]. A subset of CIN-TCPs, including LANCEOLATE (LA) from tomato, is negatively regulated by the microRNA miR319. In the tomato semi-dominant gain-of-function mutant *La*, a mutation in the miR319 binding site leads to early ectopic LA expression, resulting in precocious differentiation and small, simplified leaves [33,34,38,39]. Concurrently, premature expression of the miR319-insensitive *TCP4* in Arabidopsis plants causes early onset of maturation, resulting in a range of leaf patterning defects [37]. Downregulation of *CIN-TCP* genes by overexpression of *miR319* results in a substantial delay in leaf maturation and prolonged indeterminate growth in the leaf margin [29,33,34,36,40].

Differences in the timing of leaf growth and maturation among species and leaf positions are associated with altered *LA* expression dynamics [34]. Thus, the *LA*-miR319 balance defines the morphogenetic window at the tomato leaf margin that is required for leaf elaboration. LA activity is mediated in part by positive regulation of the hormone GA [21]. In Arabidopsis, the LA homolog TCP4 reduces CK response during leaf development [41]. Whether this effect of TCPs on CK is conserved in tomato is still unknown.

Maintenance of the morphogenetic window is also restricted by activity of the MYB transcription factor *CLAUSA* (*CLAU*) [42]. CLAU has evolved a unique role in compound-leaf species to promote an exit from the morphogenetic phase of leaf development [22]. *clau* mutants have highly compound, continuously morphogenetic leaves, in which meristematic tissues constantly generate leaflets on essentially mature leaves throughout the life of the plant [22,43]. *clau* mutants can be extremely variable in phenotype, showing that tight regulation of the morphogenetic window is also required for shape robustness [43]. CLAU regulates the morphogenetic window by attenuating cytokinin signaling and sensitivity [22]. GA application was shown to suppress the increased complexity of *clau* leaves [44], raising the possibility that CLAU also affects the GA pathway.

The tomato KNOTTED1-LIKE HOMEOBOX (KNOXI/TKN2) is a key regulator of compound leaf development. TKN2 delays leaf differentiation and preserves the meristematic identity of the leaf margin [30,45–51]. KNOXI proteins affect CK and GA levels [52–55] and both *clau* and *La* mutants show altered *TKN2* expression. These findings suggest a complex interaction among these three TFs and two central plant hormones in the regulation of the morphogenesis-differentiation balance [42,56–58].

CLAU, LA, TKN2, CK and GA were shown to modulate the morphogenetic window during tomato leaf development. However, how their activities are coordinated is not clear. In this work, we investigate the relationship between the transcription factors LA, CLAU and TKN2, and the plant hormones GA and CK, in the regulation of the morphogenesis-differentiation balance. We show that LA and CLAU effect essentially similar outcomes in tomato leaf development via likely partially parallel genetic pathways. These genetic pathways converge on the modulation of the CK/GA balance. Global transcriptomic analysis of genotypes with altered activity of LA, CLAU and TKN2 provide a molecular context by which the activity of additional regulators of morphogenesis and differentiation can be investigated.

## Results

### LA and CLAU operate in parallel genetic pathways

To better understand the genetic regulation of the balance between morphogenesis and differentiation, we examined the genetic relationships between lines with different activity levels of *CLAU* and *LA* (**Fig 1** and [22,33]. To overexpress *CLAU* specifically in leaves we used the *FIL* promoter, which is expressed from early stages of leaf development (**Fig 1E** and [48]). Increasing *CLAU* levels in a high *LA* expression background (*FIL>>CLAU La-2/+*) (**Figs 1C, 1D and 1N** and **S1**) exacerbates the highly differentiated *La-2/+* phenotype (**Fig 1B**). LA acts in a dose-dependent manner, with two mutated copies of the gain-of-function allele *La-2* promoting differentiation and reducing leaflet numbers more strongly than one mutated copy (**Fig 1B**). Interestingly, overexpressing *CLAU* from the *FIL* promoter in a heterozygous *La-2/+* background had the same effect as two mutated copies of the *La-2* allele, as demonstrated in *FIL>>CLAU La-2/+* leaves that were similar in size and complexity to *La-2* homozygous leaves (Compare **Fig 1B** and **1D**), suggesting an additive, dose-dependent interaction (**Fig 1B** and [33,39]). Decreasing *CLAU* levels in a low *LA* expression background such as the *LA* loss-of-function allele *la-6* or *FIL>>miR319* results in a significant increase in leaf elaboration

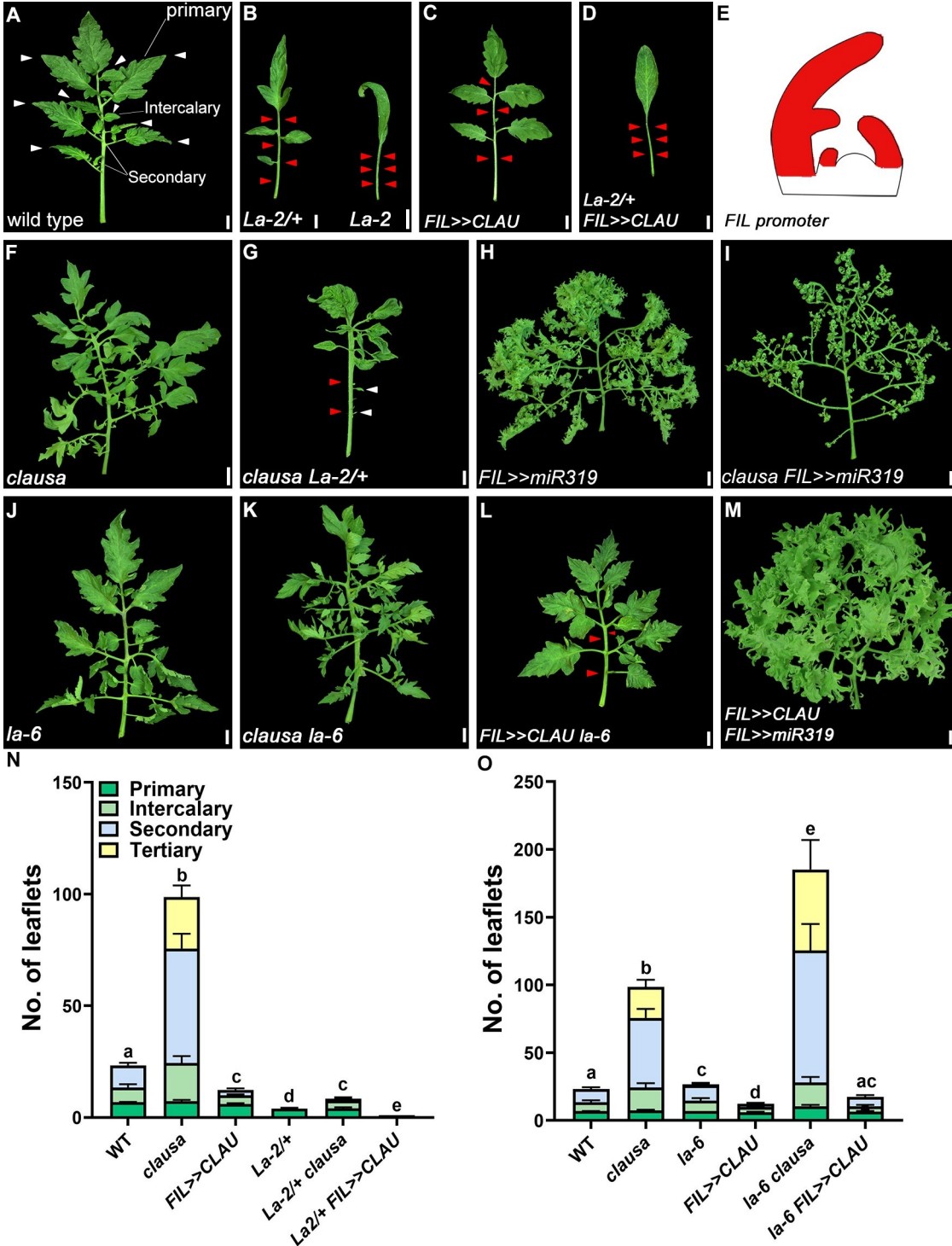

**Fig 1.** ***CLAU*** **and** ***LA*** **function in parallel pathways. (A-D, F-M)** Genetic interactions between genotypes with altered *CLAU* and *LA* expression levels. *La-2/+*: a semi-dominant LA allele with increased and precocious expression due to miR319 resistance. *La-6*, *FIL>>miR319*: *LA* null or *LA* downregulation, respectively. *clausa*: *CLAU* null. *FIL>>CLAU*: *CLAU* upregulation. All leaves depicted are fully expanded fifth leaves. Bars = 1 cm. White and red arrowheads represent primary leaflets and missing primary and intercalary leaflets, respectively. **(E)** Cartoon depicting the expression domain of the FIL promoter. **(N-O)** Quantification of leaf complexity in genotypes with altered *CLAU* and *LA* expression levels. Graphs represent mean ± SE of six independent biological repeats. Different types of leaflets are indicated according to the color code. Statistical significance of differences in the total leaflet

number was examined in a one-way ANOVA, p<0.0001. Different letters indicate significant differences between samples in an unpaired two-tailed t-test with Welch's correction. Statistical significance of differences in quantification of each leaflet type are presented in S1 Fig.

(**Figs 1F, 1H, 1I, 1J, 1K and 1O** and **S1**). Decreasing *CLAU* levels in a high *LA* expression background (**Figs 1G and 1O** and **S1**) partially rescues the highly differentiated *La-2/+* pheno-type, while increasing *CLAU* levels in a low *LA* expression background (**Figs 1L, 1M and 1O** and **S1**) results in a decrease in leaf elaboration when compared with the decreased *LA* expression genotypes. Interestingly, though the number of primary leaflets is a relatively stable trait, and deficiency in either LA or CLAU does not affect this trait, deficiency in both genes signifi-cantly increases primary leaflet number (**S1C Fig**). The effect of CLAU and LA deficiencies is additive when examining secondary and tertiary leaflet numbers (**S1E and S1F Fig**), while CLAU deficiency alone affects intercalary leaflet number (**S1D Fig**), and is not augmented by LA deficiency in this trait, suggesting that CLAU operates in the leaf rachis area while LA does not. These genetic analyses demonstrate that the effect of *CLAU* and *LA* on leaf development is partially additive, indicating that they promote differentiation in at least partially parallel pathways.

## LA activity defines the developmental window in which CLAU is expressed

Our previous results demonstrated that *LA* has a wider expression window than *CLAU*, and is active earlier in development [22,33,34]. Examination of the dynamics of *CLAU* expression in the first and fifth leaves of the plant, which represent a relatively limited and a relatively extended morphogenetic window, respectively [34], confirmed that *CLAU* is expressed mostly during the extended morphogenetic window (**S2 Fig**). As the morphogenetic window is par-tially defined by LA [33,34], this raised the possibility that low LA activity enabled the recruit-ment of CLAU in the regulation of leaf differentiation. To explore this possibility and gain more insight into the molecular basis of the additive interaction of *LA* and *CLAU* in promoting differentiation, we examined how LA activity affects *CLAU* expression, by assaying the expres-sion of *CLAU* and its promoter in successive stages of leaf development in genotypes with altered LA activity (**Fig 2**). Early maturation caused by increased *LA* expression in the gain-of-function mutant *La-2*, led to a decrease in *CLAU* expression (**Fig 2E–2H and 2M**). Conversely, delayed maturation resulting from decreased *LA* expression in *FIL>>miR319* resulted in increased *CLAU* expression (**Fig 2I–2L and 2M**). Interestingly, expressing a miR319-resistant form of LA (*op:La-2*) from the *CLAU* expression domain mimicked the *La-2/+* phenotype (**S3 Fig**), with a slightly weaker effect when compared to expressing the same *La-2* version from its own expression domain [59]. We conclude that *LA* activity defines the (spatial and temporal) developmental window in which CLAU is active. This window is reduced in *La-2/+* and is pro-longed in *FIL>>miR319*. Therefore, these TFs act in partially distinct spatial and temporal domains to promote differentiation.

## TKN2 plays an essential role in extended morphogenesis

The class I KNOX homeobox transcription factor TKN2 is a key factor promoting morpho-genesis in compound leaves [48,49,60–65]. Therefore, we set to examine the role of *TKN2* in mediating the extended morphogenesis in plants with reduced CLAU or LA activity, by com-bining them with expression of *TKN2-SRDX*, in which *TKN2* is fused to a repressive domain. Expressing *TKN2-SRDX* from the leaf-specific promoter *BLS*, which is expressed from the P4 stage of leaf development (**Fig 3H**), lacks any observable phenotype in the WT background (**Figs 3A, 3D and 3G** and **S4** and [48]), likely due to the lack of TKN2 activity at this stage in

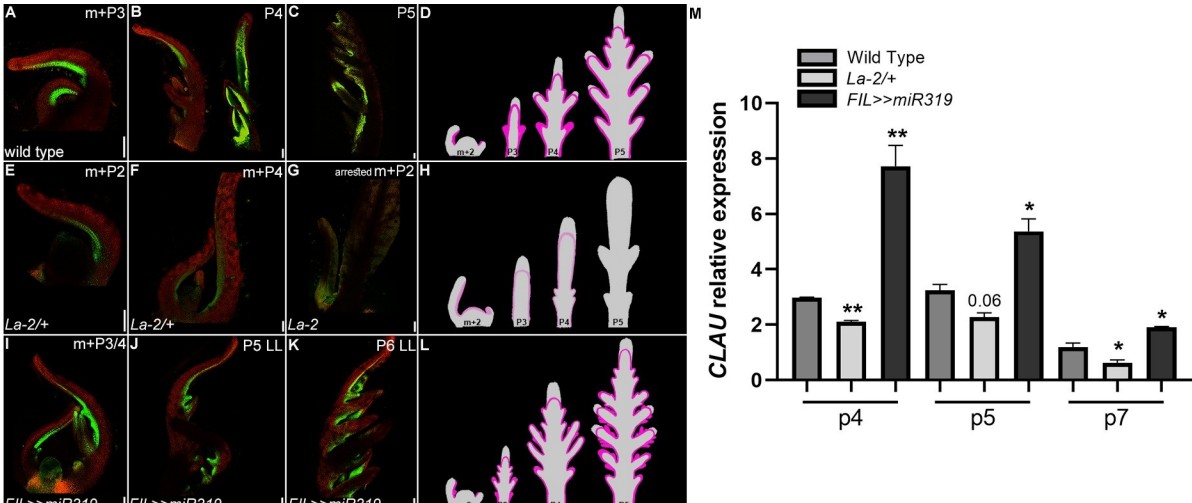

**Fig 2.** *LA* determines the developmental window in which *CLAU* is expressed. **(A-L)** Expression of the *CLAUSA* promoter *CLAU::nYFP* in altered *LA* genotypes. **(A-C)** Expression of *CLAU::nYFP* in different developmental stages of WT; **(E-G)** Expression of *CLAU::nYFP* in different developmental stages of *La-2/+* (*LA* upregulation); **(I-K)** Expression of *CLAU::nYFP* in different developmental stages of *FIL>>miR319* (Reduced *LA* activity). The pattern of *CLAU::nYFP* expression was detected by a confocal laser scanning microscope (CLSMmodel SP8; Leica), with the solid-state laser set at 514 nm excitation/ 530 nm emission. Chlorophyll expression was detected at 488nm excitation/ 700nm emission. Bars = 100 um. **(D, H, L):** Cartoon summarizing the expression of *CLAU::nYFP* throughout development in each genotype. **(M)** Expression levels of *CLAU* in altered *LA* genotypes was determined in successive leaf developmental stages using RT-qPCR. Graphs represent mean ± SE of three independent biological repeats. Asterisks indicate significant differences from *CLAU* expression in WT in an unpaired two-tailed t-test, p≤0.0387.

wild type leaves. Interestingly, *BLS>>TKN2-SRDX* suppresses the increased complexity of *CLAU* or *LA* deficient backgrounds (**Figs 3E–3G and S4**). In addition, expression of *TKN2-SRDX* from the *LA* expression domain (*LA>>TKN2-SRDX*) resulted in similar phenotypes to those of *La-2/+* mutants (**S3B Fig**). In contrast, expressing *TKN2* in either the *LA* or *CLAU* expression domains (*LA>>TKN2* and *CLAU>>TKN2*, respectively) produced highly compound leaves (**S3B–S3D Fig**). Together, these results indicate that TKN2 takes part in the increased complexity resulting from compromised CLAU and LA activities, and that *LA* and *CLAU* may both act via *TKN2*. Alternatively, TKN2 is essential for the morphogenetic stage in both the wild-type context and when this window is extended by reduced CLAU and LA activities. In agreement with previous findings [42,56], the *TKN2* promoter is more strongly activated at the leaf margin of *CLAU* or *LA* deficient backgrounds than in the wild type, while remaining mostly restricted to meristems in WT. Its expression is further elevated in the *clau la-6* double mutant background (**S5 Fig**). This effect can again be an effect of the extended morphogenetic window and of *TKN2* expression being one of the molecular characteristics of this window.

To further investigate the functional interaction among these factors, we examined the effect of combining altered *CLAU* and *LA* expression with *TKN2* overexpression (**Fig 4**). Overexpressing *TKN2* (**Fig 4G**) in a *CLAU* (**Fig 4B**) or *LA* (**Fig 4D and 4E**) deficient background (**Fig 4H, 4J, 4K and 4N**), leads to highly compound leaf forms. Overexpressing *TKN2* in a *CLAU* (**Fig 4C**) or *LA* (**Fig 4F**) overexpression background (**Fig 4I, 4L and 4N**) leads to increased relative leaf elaboration and a rescue of the simplified leaf forms generated by overexpression of *CLAU* or *LA*. This rescue is substantial in the case of *CLAU*, and more moderate in the case of *La-2/+* (**Fig 4**). Terminal leaflets are exemplified in shading in **Fig 4M**. These results indicate that the phenotypes observed upon loss of function of *CLAU* or *LA* are not solely due to *TKN2* and that there are other morphogenesis-differentiation processes mediated

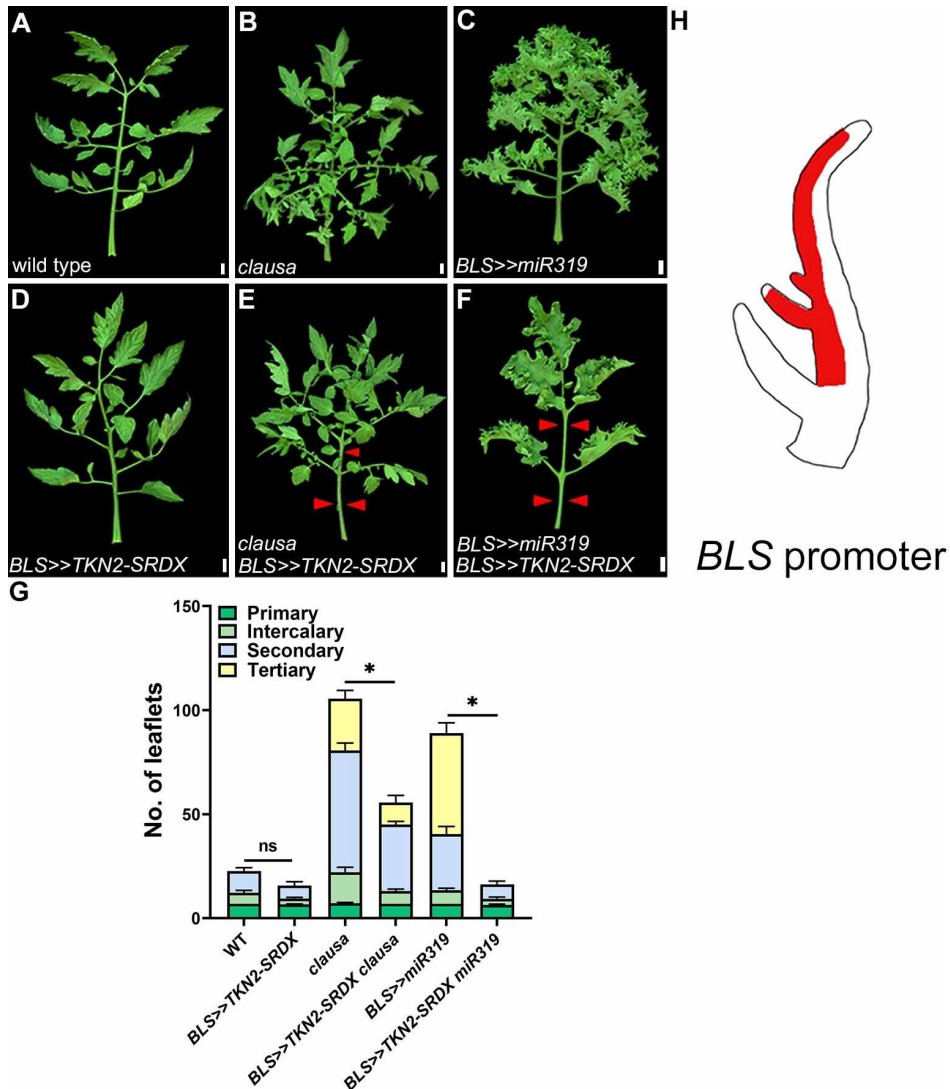

**Fig 3. TKN2 mediates the increased morphogenesis in *CLAU*- and *LA*-deficient backgrounds. (A-F)**
Overexpression of *TKN2-SRDX* in the background of *CLAU* and *LA* deficiency. All leaves depicted are fully expanded
fifth leaves. Bars = 1 cm. Red arrowheads represent primary leaflets and missing primary and intercalary leaflets. **(G)**
Quantification of leaf complexity upon overexpression of TKN2-SRDX in the background of *CLAU* and *LA* deficiency.
Graphs represent mean ± SE of at least three independent biological repeats. Different types of leaflets are indicated
according to the color code. Asterisks indicate significant differences of the total leaflet number from the background
genotype (without TKN2-SRDX overexpression) in an unpaired two-tailed t-test with Welch's correction, p≤0.0335.
Statistical significance of differences in quantification of each leaflet type are presented in S4 Fig. **(H)** Cartoon
depicting the expression domain of the BLS promoter.

by LA and CLAU. Overall, these results suggest that TKN2 acts antagonistically to CLAU and
LA in tuning the morphogenesis-differentiation balance, and that reduced CLAU and LA
activities enable the extension of morphogenesis, in which TKN2 plays a central role.

## CLAU and LA converge on the CK-GA balance

We hypothesized that LA, CLAU and TKN2 may converge on common downstream pathways
that are involved in leaf development. Several evidences suggest that these TFs regulate the

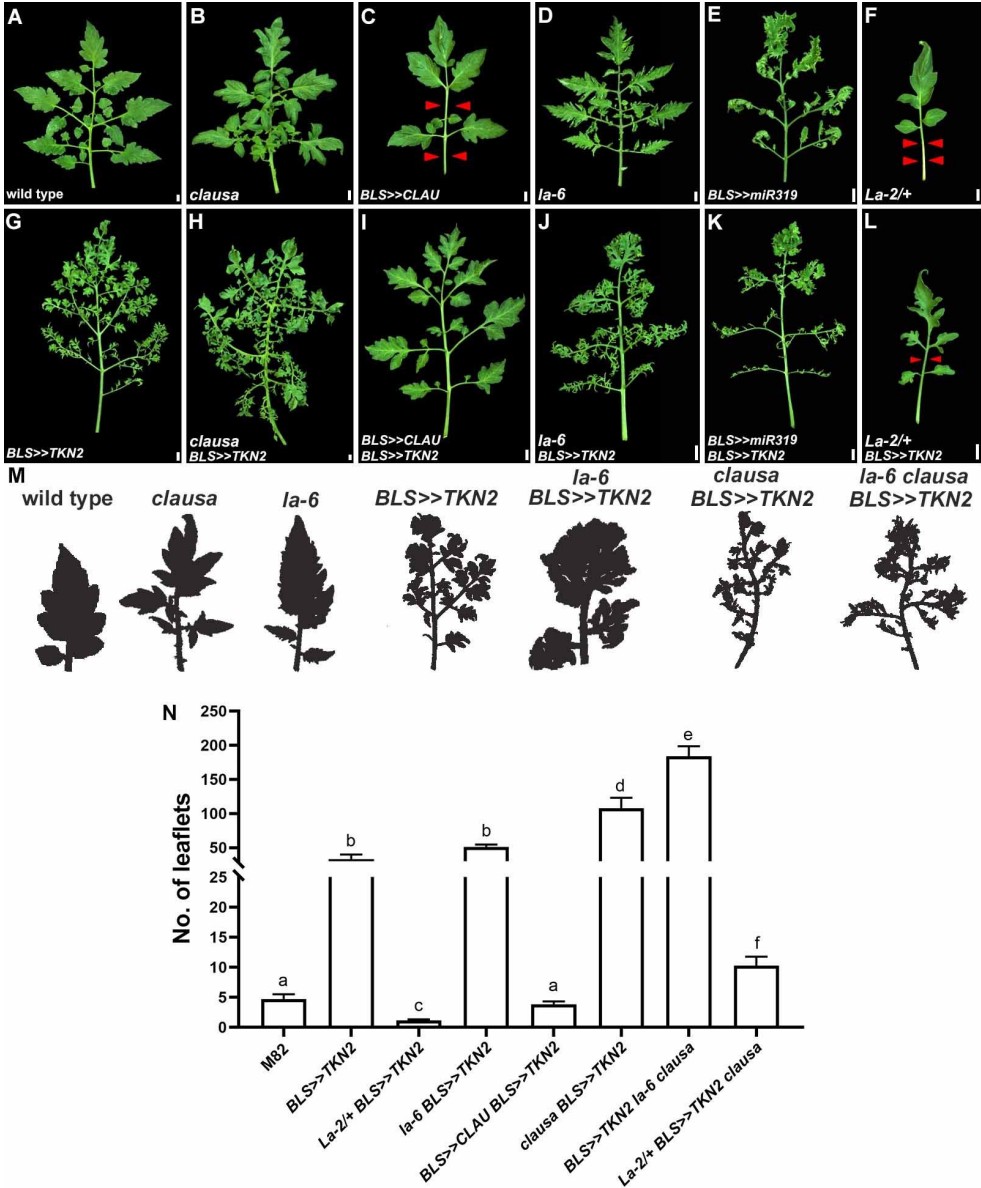

**Fig 4. TKN2 promotes morphogenetic activity in altered *CLAU* and *LA* backgrounds. (A-L)** Overexpression of *TKN2* in the background of genotypes with altered *CLAU* and *LA* expression levels. All leaves depicted are fully expanded fifth leaves. Bars = 1 cm. Red arrowheads represent missing primary and intercalary leaflets. **(M)** Shaded cartoon of the terminal leaflet of the indicated genotypes. **(N)** Quantification of leaf complexity of the second left-hand lateral leaflet of leaf No. 5 upon overexpression of *TKN2* in genotypes with altered *CLAU* and *LA* expression levels. Graphs represent mean ± SE of at least three independent biological repeats. Letters indicate significant differences between samples in a one-way ANOVA with a Tukey post-hoc test, p<0.027.

balance between the two plant hormones GA and CK. We previously demonstrated that *CLAU* functions through attenuation of CK signaling [22]. We and others have also previously shown that LA functions in part through GA signaling [21,66]. Previous work has also demonstrated that the Arabidopsis *LA* homolog, *TCP4*, reduces leaf CK response through binding and promoting expression of the CK response inhibitor *ARR16* [41]. TKN2 was shown to affect both GA and CK [52,53,64]. Since both CLAU and LA promote differentiation in

different pathways and spatiotemporal windows, and since CK and GA are partially antagonistic in leaf development [22,27,33,67], we examined the relationship between *CLAU* and GA, and *LA* and CK. In agreement with the antagonistic relationship between CK and GA in leaf morphogenesis, reducing CK content by overexpression of the CK inactivation gene *CKX*, or application of GA, led to simplification of leaf form (**Figs 5B, 5D and 5N** and **S6**), and combining reduced CK with increased GA further reduced leaf complexity (**Figs 5G and 5N** and **S6**). Interestingly, the leaves of simultaneously reduced CK and increased GA levels resulted in phenotypes that were very similar to that of *La-2* and *CLAU* overexpressing plants (**Fig 1**). Conversely, inhibition of GA response via overexpression of a GA-resistant form of the GA response inhibitor DELLA/PROCERA (PRO) (PROΔ17) results in increased leaf complexity (**Fig 5E**). Interestingly, *FIL>>PROΔ17* had an increased number of intercalary and secondary leaflets, similar to *clausa* mutants (**Figs 1F** and **S1B** and **S1D**), suggesting that CLAU could act via GA. In agreement, the simplified leaf phenotype caused by *CLAU* overexpression (**Fig 5C**) is rescued by co-expression of *PROΔ17* (**Figs 5F and 5N** and **S6**), suggesting that GA may mediate the effect of *CLAU* on leaf differentiation.

These results suggested that *CLAU* may act via increased GA activity in addition to its inhibition of CK response, and that LA may act on the CK/GA balance also by reducing CK, as has been shown in Arabidopsis [41]. To test this hypothesis, we examined the effect *clausa* on GA content. Interestingly, GA4 and GA20 amounts were substantially reduced in 14-day-old *clausa* shoot apices, while the content of the more upstream GAs GA53 and GA19 increased (**Fig 5K**). This demonstrates that the GA pathway is blocked in *clausa* mutants at the step of the conversion of GA19 to GA20, a step catalyzed by GA20ox. In agreement with the accumulation of GA19 and decrease in GA20, and with previous findings [44], the expression of the tomato GA biosynthesis gene *GA20oxidase-1* (*SlGA20ox-1)* was reduced in *clausa* mutants (**Fig 5L**). These results suggest that CLAU promotes differentiation by regulating GA biosynthesis, and that in *clau* mutants, reduced levels of the GAs GA20 and GA4 and/or GA response facilitate prolonged morphogenesis and compound leaf shape.

To examine whether CLAU also influences the leaf sensitivity to GA, we treated WT and *clausa* plants with increasing GA concentrations (**Figs 5H, 5I, 5J and 5O** and **S7**). The *clausa* mutant displayed a strong and significant reduction in GA sensitivity at the leaf margin, remaining highly compound despite GA treatments at WT-responsive concentrations (0.01–1 uM GA), and responding only to a whopping 10 uM of GA (**Figs 5O** and **S7**). Therefore, CLAU exerts its role in regulating differentiation through regulation of both GA levels and response.

In kind, The *La-2/+* simple-leaf phenotype is exacerbated by overexpression of the CK inactivation gene *CKX)La-2/+ FIL>>CKX)* (**Fig 6B**). The reduced CK levels resulting from *CKX* overexpression were phenotypically equivalent to an extra mutated *La-2* copy, similar to the effect of *CLAU* overexpression in a *La-2/+* background (compare **Figs 1B and 1D** and **6B**) [22]. In agreement, *La-2/+* is partially rescued by overexpression of CK biosynthesis gene *IPT* (*La-2/+ FIL>>IPT*) (**Fig 6D**). Reducing CK in a *LA* deficient background shortens the morphogenetic window, partially rescuing the super compound phenotype of *FIL>>miR319* (**Fig 6F**). In addition, similar to the arabidopsis TCP4, we found that LA reduced leaf sensitivity to CK, as evident from both the reduced signal of VENUS driven by the synthetic CK responsive promoter *TCSv2* [68] in *La-2/+* primordia (**Figs 6G–6I** and **S8**), and the increase in CK-dependent anthocyanin accumulation in LA deficient plants. As expected, in WT leaf primordia, the mean *TCSv2* signal decreases during development (**S8G Fig**), as the CK signal decreases, while the corrected total fluorescence increases (**S8H Fig**), because the area being measured is larger as the primordia grows. We observed no differences in the fluorescence level in the meristem of the different genotypes (**S8G and S8H Fig**), and a reduction in *TCSv2*

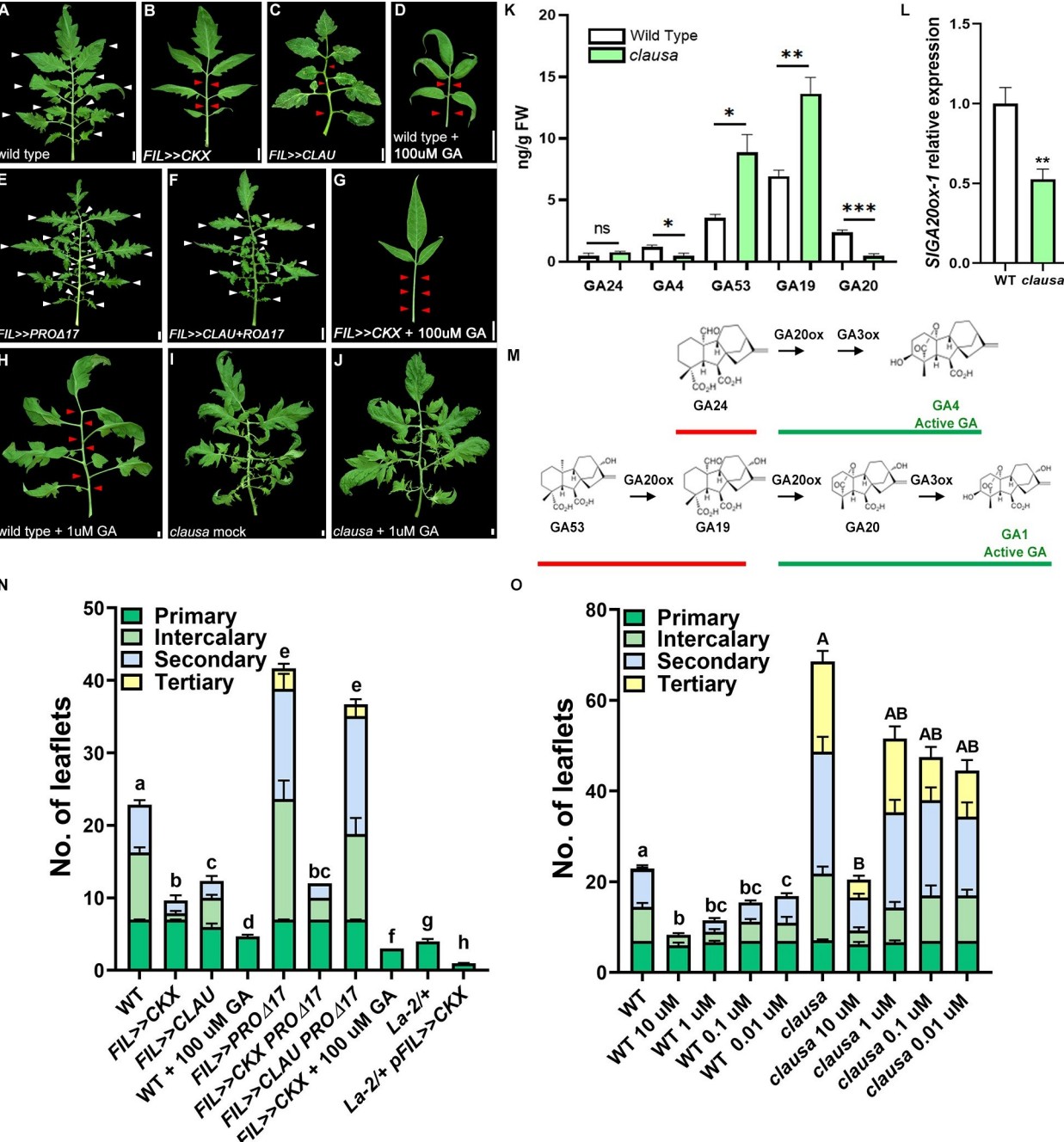

**Fig 5. *clausa* has an altered GA profile and reduced sensitivity to GA treatment. (A-D)** Phenotypes of leaves with reduced CK (*FIL>>CKX*) increased CLAU (*FIL>>CLAU*), or that received exogenous GA. **(E-G)** Phenotypes of leaves with reduced GA response (*FIL>>PROΔ17*), or increased CLAU and reduced GA response (*FIL>>CLAU+ PROΔ17*), or reduced CK content (*FIL>>CKX*) treated with GA. **(H-J)** Effect of GA treatment on WT and *clausa*. All leaves depicted are fully expanded fifth leaves. Bars = 1 cm. White and red arrowheads represent primary leaflets and missing primary and intercalary leaflets, respectively. **(K)** Quantification of GAs in WT and *clausa*. Asterisks indicate significant differences between WT and *clausa* for each GA in an unpaired two-tailed t-test, *p≤0.05, **p≤0.01, ***p≤0.001. **(L)** Expression of *SlGA20ox-1*, the enzyme that converts GA19 to GA20, in WT and *clausa*, was determined in young shoots (m+6) of 2-week-old plants using RT-qPCR. Graphs represent mean ± SE of five independent biological repeats. Asterisks indicate significant differences in an unpaired two-tailed t-test, p = 0.0078. **(M)** Depiction of GA biosynthesis pathways arrested in *clausa*. The compounds GA24, GA53 and GA19 accumulate (red underline) while the active GA4 and the GA1 precursor GA20 (green underline) are reduced, suggesting reduced GA20ox function. **(N)** Quantification of leaf complexity. Graphs represent mean ± SE of at least three independent biological repeats. Different types of leaflets are indicated according to the color code. Statistical significance of differences in the total leaflet number was examined in a one-way ANOVA, p<0.0001. Different letters indicate significant differences between samples in an unpaired two-tailed t-test with Welch's correction (p<0.015). Statistical significance of differences in

quantification of each leaflet type are presented in S5 Fig. **(O)** Quantification of leaf complexity following GA treatment in WT and *clausa*. Graphs represent mean ± SE of at least three independent biological repeats. Different types of leaflets are indicated according to the color code. Statistical significance of differences in the total leaflet number was examined in a one-way ANOVA, p<0.0001. Different letters indicate significant differences between samples in a Dunnett post-hoc test (p<0.0097). Statistical significance of differences in quantification of each leaflet type are presented in S6 Fig.

driven expression in *La-2/+* leaf primordia compared with WT, in both mean fluorescence (from the P2 stage) and corrected total fluorescence (from the P3 stage). CK promotes antho-cyanin accumulation [27,41,69]. We therefore measured anthocyanin levels in the different LA genotypes as an additional measure of the effect of LA activity on CK sensitivity. LA deficiency caused increases in CK dependent anthocyanin accumulation (**S9 Fig**). LA was found to bind *in vitro* to the promoters of tomato response regulators (A-type TRRs) involved in CK signaling (**S10 Fig**). We found that LA, CLAU and the CK/GA balance also affect inflorescence complexity in a similar manner to their effect in leaves (**S11 Fig**). Thus, we conclude that both CLAU and LA enhance differentiation by reducing the plant's sensitivity to CK and by elevating GA levels and/or response. Together, LA and CLAU affect the GA/CK balance, in turn tuning the morphogenesis-differentiation balance.

## Global transcriptomic approach to identify common molecular pathways of morphogenesis and differentiation

To gain insights on leaf morphogenesis at the molecular level, we compared global transcriptomic data among the genotypes included in this study. Our findings suggest that the key regulators: LA, CLAU and TKN2 act in partially parallel pathways but also converge on the same downstream processes in the regulation of the balance between morphogenesis and differentiation. We thus compared several transcriptomic data sets from various genetic backgrounds with different activity of CLAU (the meristem and the four youngest leaf primordia of WT vs *clausa*) [22], LA (the meristem and the two youngest leaf primordia)-of *La-2/+* gain-of-function), WT, *la-6* loss-of-function, and *FIL>>miR319* that down regulates *LA* and three additional *CIN-TCPs*: *TCP3*, *TCP10* and *TCP24*) [59] and TKN2 (the meristem and the five youngest leaf primordia of *BLS>>TKN2* vs WT and *BLS>>TKN2-SRDX*) [48] (**Fig 7**). Micro-array data sets for the *LA* genotypes and *TKN2*, and RNAseq data for the *clausa* mutant, were analyzed for Fold change. All data sets were generated by the Ori group, using the M82 background, with plants grown under essentially similar conditions in a controlled growth chamber. KEGG (Kyoto Encyclopedia of Genes and Genomes) analysis was conducted to identify significantly differential pathways. Each genotype was compared to the M82, wild type background for the analysis. Differentially expressed genes (DEGs) confirm dependencies between the LA genotypes, with between about a third to half of the genes significantly upregulated in *La-2/+* being significantly downregulated in *la-6* and upon *miR319* overexpression (**S1 Data**). Likewise, about a third of the genes significantly downregulated in *La-2/+* are significantly upregulated in *la-6* or upon *miR319* overexpression (**S1 and S2 Datas**).

Interestingly, commonly up/downregulated genes are overrepresented between LA data-sets and TKN2 datasets, with 2–3 times more DEGs than expected being commonly upregulated in *La-2/+* and downregulated upon *TKN2* overexpression or upregulated upon *TKN2-SRDX* overexpression (**S1 and S2 Datas**). In agreement with our genetic and molecular analyses, genes upregulated upon low *LA* expression (*la-6*, *miR319* overexpression), or genes downregulated upon high *LA* expression (*La-2/+*), correlate best with those upregulated upon *TKN2* overexpression or downregulated upon *TKN2-SRDX* overexpression (**Fig 7A and 7B**). This demonstrates that, to a degree that is significantly higher than expected

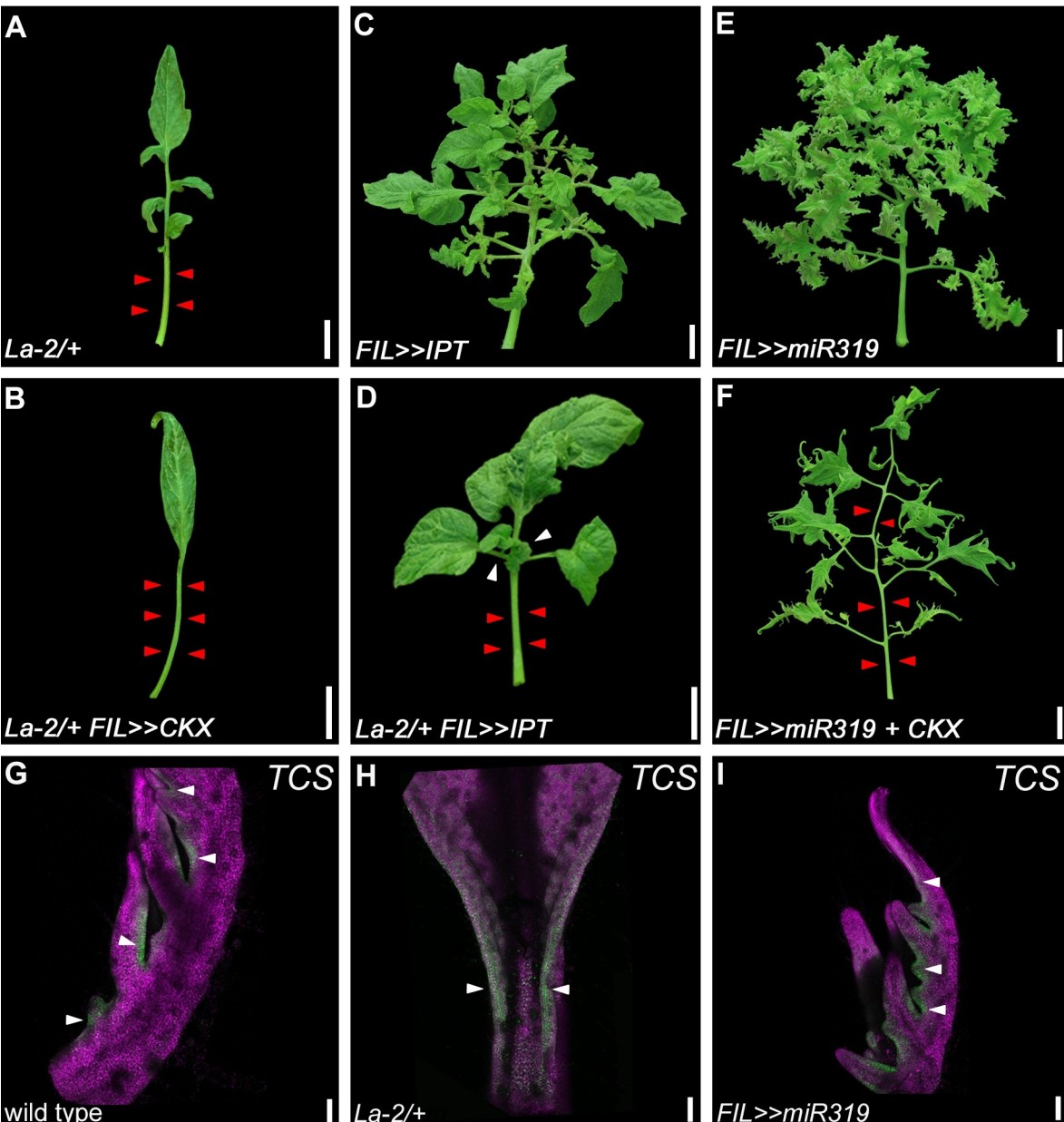

**Fig 6.** *LA reduces leaf margin's sensitivity to CK.* (**A-F**) Phenotypes of leaves with increased CK (*FIL>>IPT*) or decreased CK (*FIL>>CKX*) levels, and increased *LA* activity (*La-2/+*) or reduced *LA* activity (*FIL>>miR319*). All leaves depicted are fully expanded fifth leaves. Bars = 2 cm. White and red arrowheads represent primary leaflets and missing primary and intercalary leaflets, respectively. (**G-I**) Confocal micrographs of *TCSv2::3XVENUS* in successive developmental stages in indicated genotypes. *TCSv2* driven signals are reduced in *La2/+* in the fifth plastochron. The pattern of VENUS expression was detected by a confocal laser scanning microscope (CLSM model SP8; Leica), with the solid-state laser set at 514 nm for excitation and 530 nm for emission. Chlorophyll expression was detected at 488nm excitation/ 700nm emission. Bars = 100 um.

from random sampling, the extended morphogenesis upon absence of LA activity correlates with increased TKN2 activity, and with downregulation of processes which are affected by inhibition of TKN targets.

KEGG pathway analysis of DEGs in all samples revealed that, in agreement with the results of this study (**Figs 5 and 6**) and with published data [21,22], plant hormone signal

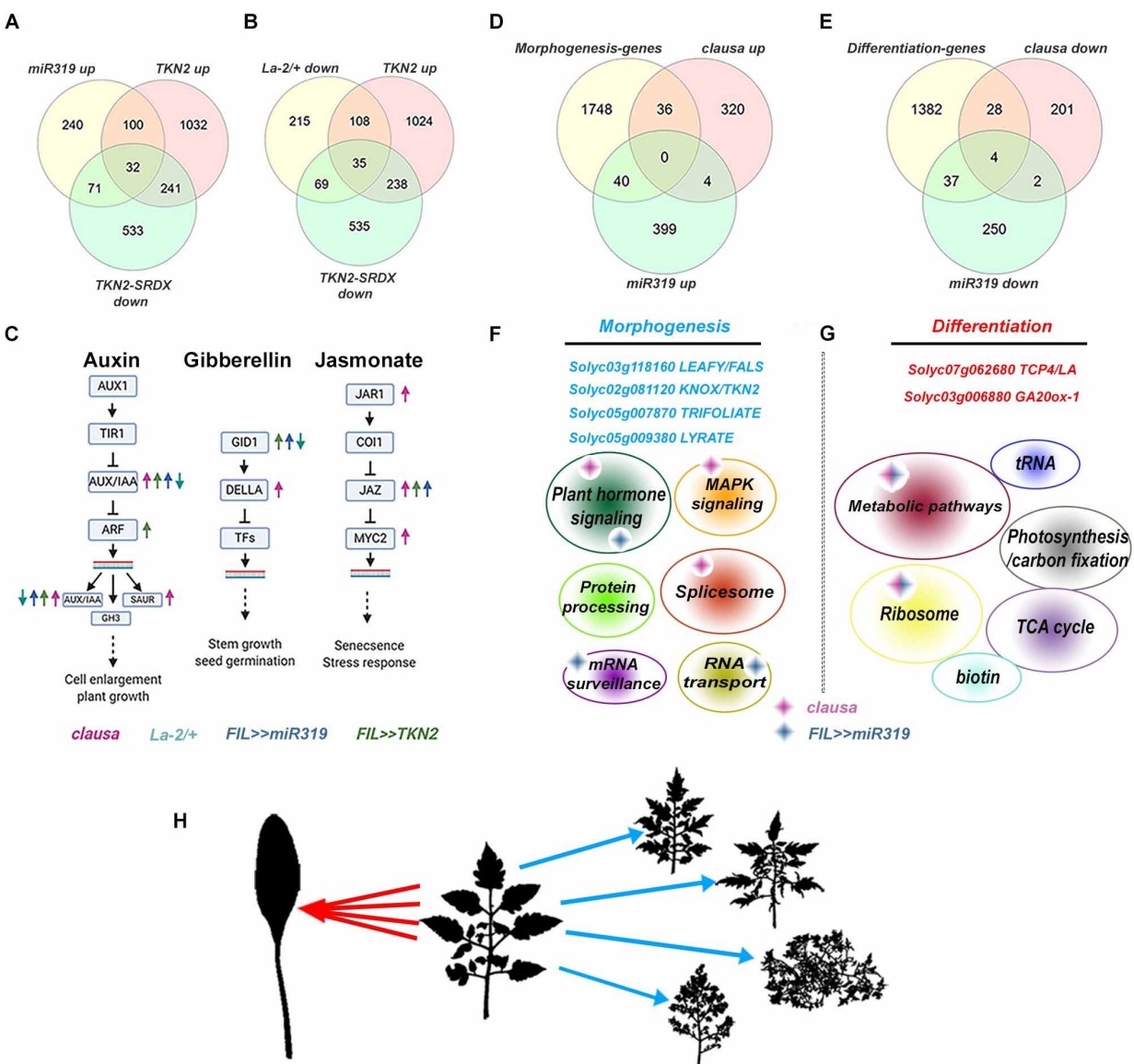

**Fig 7. Global expression analyses of LA, CLAU and TKN2.** (**A-B**) Venn diagrams depicting co-regulated genes in LA and TKN2 genotypes. Roughly a third to half of the genes upregulated upon LA activity downregulation (*miR319* overexpression), or downregulated upon LA activity upregulation (*La-2/+*), are co-regulated upon upregulation of TKN2 activity (overexpression of *TKN2*) and/or downregulation of the activity of TKN2 targets (overexpression of *TKN2-SRDX*). (**C**) Hormone signaling pathways co-regulated in the different genotypes. Arrow color corresponds with genotypes provided below, arrow direction indicates up or down regulation. (**D-G**) CLAU and LA promote differentiation in partially discrete pathways. Morphogenesis and differentiation genes were identified from a published dataset (Ichihashi et al., 2014). 1824 genes fit the condition of m+P3 exp> P4 exp> P5 exp> P6 exp and are considered "Morphogenesis" genes. 1451 genes fit the condition of m+P3 exp< P4 exp< P5 exp< P6 exp and are considered "Differentiation" genes. (**D**) "Morphogenetic" genes, as a group, are significantly enriched in the upregulated group in *clausa* (p<0.014) and *pFIL>>miR319* (p<0.044). (**F**) Known morphogenesis genes identified that validate the methodology are listed in blue. Cartoon depicts the pathways significantly enriched in morphogenesis genes, denoting significant upregulation in *clausa* (pink diamond) and *FIL>>miR319* (blue diamond). None of the genes overlap, and both common and distinct pathways are represented in the two genotypes. Fisher's exact test was used to calculate statistical significance of KEGG pathway enrichment; p-values were corrected by calculating false discovery rate (FDR) using the Benjamini Hochberg test, corrected p-value<0.05. (**E**) "Differentiation" genes, as a group, are significantly enriched in the downregulated group in *clausa* (p<0.0001) and *pFIL>>miR319* (p<0.0001). (**G**) Known differentiation genes identified that validate the methodology are listed in red.—Cartoon depicts the pathways significantly enriched in differentiation genes, denoting significant downregulation in *clausa* (pink diamond) and *pFIL>>miR319* (blue diamond). Only common pathways are represented in the two genotypes, and 10–15% of the genes overlap. Fisher's exact test was used to calculate statistical significance of KEGG pathway enrichment; p-values were corrected by calculating FDR using the Banjamini Hochberg test, corrected p-value<0.05. (**H**) Cartoon depicting morphogenesis and differentiation pathways. Different morphogenetic genes generate different pathways (blue arrows) that result in different patterning with similar leaf complexity. Different as well as overlapping differentiation genes generate different, partially overlapping pathways (red arrows) that, in the most simple leaved, differentiated form, look similar in addition to having equal complexity. See also Figs 8 and S1, S4 and S6 and S3 and S4.

transduction pathways are affected in *TKN2*, *CLAU* and *LA* genotypes (**Fig 7C and S1 and S2 Datas**). For example, GA signaling is altered in these genotypes, with *DELLA/PROCERA* upregulated in *clausa* and the GA-receptor *GID1* upregulated in *miR319* and *TKN2* and downregulated in *La-2/+* and *TKN2-SRDX*, providing molecular context for the altered sensitivity of these genotypes to GA. Interestingly, jasmonate pathways are upregulated in all the "moprphogenetic" genotypes, most strongly in *clausa*, and ethylene signaling is uniquely upregulated in *clausa*, in addition to upregulation of plant pathogen interaction (p = 0.00058) and MAPK signaling (p = 0.0096) pathways (**S1 and S2 Datas**). We have previously demonstrated that *clausa* is immuno-active and pathogen resistant [70]. Direct comparisons of the CLAU data generated by RNAseq, to the LA and TKN2 data sets, generated by microarray hybridization, was not undertaken in depth due to the methodological differences.

Analysis of increased morphogenesis genotypes (*la-6*, *miR319* overexpression, *clausa* and *TKN2* overexpression) revealed a significant increase in metabolic processes, carbon fixation, biosynthesis of amino acids and glycolysis—perhaps required for the increase in activities required for leaflet generation (**S1 and S2 Datas**). Furthermore, it emerges that *LA* and *TKN2* co-regulate protein processing and protein modification, with pathways of ER protein processing being upregulated in *la-6* and *TKN2* and downregulated in *La-2/+*, and Glycosylphosphatidylinositol (GPI)-anchor biosynthesis being upregulated in *la-6* and *miR319* overexpression, and down regulated in *La-2/+* and *TKN2-SRDX* (**S1 and S2 Datas**).

We compared our transcriptomic data to published data [71] that includes three solanum species at four developmental stages. In the public data we focused on genes that showed successive downregulation throughout the developmental stages in the M82 background, which we termed 'morphogenesis genes', and on genes that were successively upregulated throughout the developmental stages in the M82 background, to whom we referred to as 'differentiation genes'. Many classical morphogenesis genes were identified in the dataset based on this rule, validating our approach (see **Fig 7F and 7G and S3 Data**). When comparing these sets of 'morphogenesis' and 'differentiation' genes with the DEGs in our genotypes (**Fig 7D and 7E**) we found that, in nearly all cases, morphogenesis genes were significantly enriched in the morphogenetic genotypes *clausa*, *la-6*, and overexpression of *miR319* or *TKN2*, while differentiation genes were significantly depleted in these genotypes (**Fig 7 and S3 and S4 Datas**). In agreement, differentiation genes were significantly enriched in *La-2/+* and depleted in the morphogenetic genotypes (**S3 and S4 Datas**). Interestingly, the morphogenetic genes upregulated in *clausa* and *miR319* overexpression showed no overlap, while the differentiation genes depleted in *clausa* and *miR319* overexpression showed 10–15% overlap, supporting the hypothesis that emerges from our results, that *CLAU* and *LA* may regulate different genetic pathways in the leaf developmental program (**Fig 7**). While the absence of CLAU or LA activity can result in highly compound leaf forms, which may be similar in leaflet number (**Figs 1,3 and 4**), their patterning results in visibly different leaf forms. This can be a result of spatial differences in their expression (**S2 and S3 Figs**) as well as their participation in different downstream pathways, as evidenced by their genetic interaction (**Fig 1**) and their influence on different "morphogenetic" genes (**Fig 7D, 7F and 7H**). While there are several patterns in which a leaf can be compound, the most simplified leaf forms generally resemble each other (**Fig 7H**).

## Discussion

Investigating the underlying molecular mechanism of shape formation is crucial for our understanding of organ function. In this work, we used partially elucidated pathways to probe the convergence of two important transcription factor developmental regulators, CLAU and LA, on the balance between CK and GA, from different angles. We examined how the key

regulators LA, CLAU and TKN2 interact in promoting and tuning differentiation during leaf development. Analysis of the interaction between *CLAU* and *LA* indicates that they operate in mostly parallel pathways (**Figs 1 and S1**), as LA exacerbates the effect of CLAU deficiency, leading to super-elaboration, while CLAU overexpression rescues LA deficiency, and LA over-expression rescues CLAU deficiency. This is also evident from the bioinformatics analyses (**Fig 7 and S3 and S4 Datas**). Our work suggests that LA might determine the developmental window during which CLAU acts (**S2 Fig**) [34]. Thus, despite the mostly parallel genetic pathways, decreased or delayed LA expression during early wild-type leaf development enables the existence of an extended morphogenesis window during which *CLAU* is expressed. CLAU in turn restricts further expansion of this window. Precocious LA activity thus renders CLAU activity irrelevant, since the spatio-temporal domain in which CLAU is active is essentially abolished upon an early increase in LA activity. This is in agreement with the unique role of CLAU in compound leaf species [22]. It will be interesting to identify additional compound-leaf specific regulators that were recruited in the context of extended morphogenesis. Similarly, the class I KNOX homeobox transcription factor TKN2 was also investigated in the context of extended morphogenesis of compound leaves [48,49,63–65], and was shown here to partially mediate the effect of both LA and CLAU in the regulation of the morphogenesis-differentiation balance. Here, we show that the CK-GA balance is a common mechanism that mediates both LA and CLAU activity. Leaf development has been reported to depend on the balance between CK, which promotes morphogenesis, and GA, which promotes differentiation [27,52,53,64,72,73]. The genetic interaction shown here between LA and CK (**Fig 6**) suggests that LA acts in part by reducing CK sensitivity. This is also supported by a previous report showing an effect of TCP on the sensitivity to CK [41], suggesting that the role of TCP4 and LA is conserved between Arabidopsis and tomato. LA differentiation-promoting activity was shown to also depend on GA response [21,66,74]. CLAU promotes differentiation by elevating GA levels, and, in its absence, the plant becomes less sensitive to GA treatment at the leaf margin (**Fig 5**). We previously demonstrated that CLAU reduces CK sensitivity [22]. It thus emerges that CLAU and LA converge on the CK-GA balance: both promote differentiation by increasing the plants' response to GA and reducing its response to CK. The length of the morphogenetic window within leaf differentiation can thus be viewed as an almost binary "lever" of sorts: pulling the lever towards CK will lengthen the window, while pulling it towards GA will shorten the window. It seems that the differentiation-morphogenesis and CK-GA balances are regulated and interpreted in a dose dependent manner (**Fig 8**). The mutation in the miR319 recognition site in *La-2* is dominant, with the homozygote being more severely affected than the heterozygote (**Fig 1B**) [33,74]. Our results demonstrate a "dose" effect of transcription factor activity and hormone levels that is translated to leaflet number. Overexpression of both CLAU and LA, or either one of these transcription factors overexpressed with CKX (**Figs 1 and 6**; [22]), or the homozygous version of the dominant *La-2* mutant, all exhibit simple leaves without any leaflets, indicating that the capacity for morphogenesis is embodied in the activity of LA, CLAU, CK and GA, acting in concert. It may suggest that additional regulators that were co-opted into the developmental program of compound leaves are regulating this balance. For example, KNOXI proteins such as *TKN2* regulate the CK-GA balance, by negatively regulating the expression of the GA biosynthesis gene *GA20oxidase* (*GA20ox*) and positively regulating the GA deactivation gene *GA2oxidase* (*GA2ox*) [52,54,55,64]. KNOXI proteins also activate CK biosynthesis genes and promote CK accumulation [52,53,55]. Here we show that *GA20ox-1* is positively regulated by CLAU (**Fig 5**). It is therefore possible that the regulation of the CK-GA balance by CLAU and LA may be mediated in part through pathways common with *TKN2*. The GA-CK balance also plays a key role in meristem maintenance,

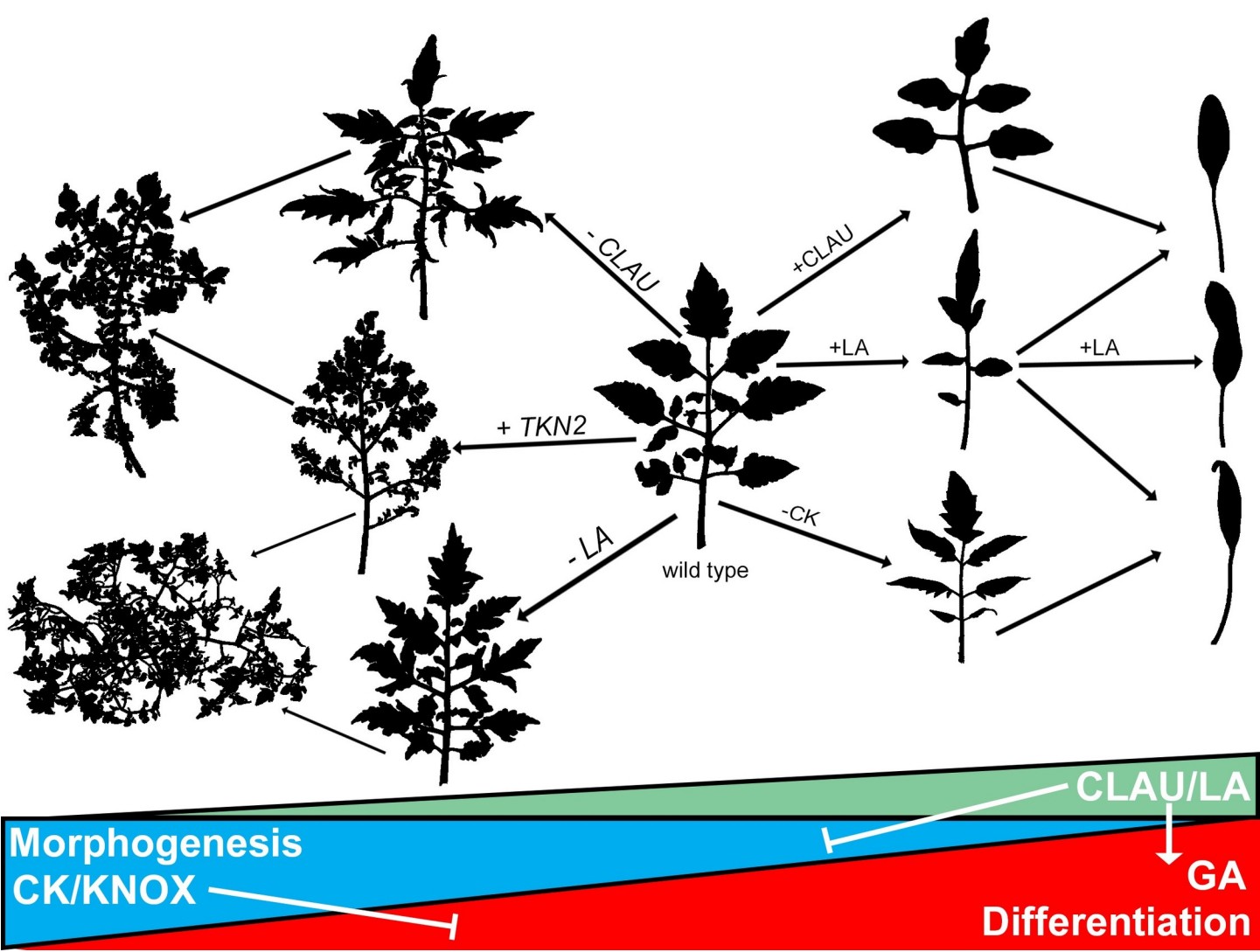

**Fig 8. Model for the role of *CLAU, LA and TKN2* and GA-CK balance in leaf development.** CK promotes morphogenesis, and GA promotes differentiation. Increasing CLAU or LA levels leads to increased GA sensitivity and decreased CK sensitivity, pulls the leaf developmental program towards differentiation, and results in simplified leaf forms. Decreasing CLAU or LA levels leads to increased CK sensitivity and decreased GA sensitivity, pulls the leaf developmental program towards morphogenesis and results in elaborate leaf forms. LA acts in a dose-dependent manner. While only 1 copy of the dominant LA allele in *La-2/+* gain-of-function mutants produces some leaflets, adding additional copy of LA allele in the *La-2* homozygous plants produces a completely simple leaf. The additional LA copy can be replaced by increasing CLAU levels, by reducing CK levels or by increasing GA levels [39]. Reducing CLAU or LA activity or increasing TKN2 activity leads to increase in leaf complexity, which is further enhanced when combining two of them.

which highlights the similarities between the shoot apical meristem and the transient meristematic phase in the leaf primordia that is required to preserve and enable organogenesis [75].

**Fig 8** details a model depicting the roles of CLAU, LA and TKN2 in the CK/GA balance during leaf development. Both LA and CLAU may promote differentiation via inhibition of TKN2, though they also appear to have TKN2 independent activity. The activity of different transcription factors may affect the location of the lever between CK and GA and can do so within different spatial-temporal domains of the developmental program.

We conclude that CLAU and LA operate in mostly parallel pathways, having different spatio temporal expression patterns and likely different downstream targets, based on our transcriptomic analyses. Their activity "funnels" from different pathways into the CK-GA balance,

which is a common downstream mechanism to regulate morphogenesis and differentiation in plants. Overall, the genetic, molecular, and transcriptomic analyses we present here, provide insights into the molecular basis of differentiation and morphogenesis processes in plants. Our transcriptomic approach and generated data could be a potential starting point to identify and examine additional potential morphogenesis and differentiation factors in additional species and developmental processes.

## Materials and methods

### Plant material

Tomato seeds (*Solanum lycopersicum* cv M82 or as indicated) were sown in a commercial nursery and grown in the field or in a glasshouse under natural daylight with 25:18°C (day: night) temperatures and a maximum light intensity of 450 μmol m$^{-2}$ s$^{-1}$. For developmental and expression analyses, plants were grown in a controlled growth chamber, 300 μmol m$^{-2}$ s$^{-1}$ 18 h/6-h light/dark regime. All complexity analyses were conducted on leaf No. 5 of 7–8 week old plants.

Genotypes used in the present study were previously described: *clausa* [22,43,76]. *pFIL>>-CLAU* [22]. *La-2/+* and *pFIL>>miR319* [33,34]. *pFIL>>IPT* and *pFIL>>CKX* [20]. *pBLS>>TKN2 and pBLS>>TKN2-SRDX* [48]. *pFIL>>PROΔ17* [77]. *pTKN2::nYFP* was generated by amplifying ~5500 bp of genomic DNA upstream to the tomato *TKN2* atg using the primers detailed in S1 Table, fusing them to YFP with a nuclear localization signal, and transforming tomato plants–essentially as previously described for *pCLAU::nYFP* [22]. Additional genotypes were generated by crossing these genotypes, where indicated. *pTKN2::nYFP*, *pCLAU::nYFP* [22], and *pTCSv2:3XVENUS* [22,78,79], were backcrossed into the relevant backgrounds.

### Tissue collection and RNA analysis

Tissue collection, RNA preparation, and qRT-PCR analysis were performed as previously described [34]. Expression of all assayed genes was normalized relative to tomato EXPRESSED (EXP). Primer sequences used in qRT-PCR analyses are detailed in S1 Table.

### Imaging

Leaves were photographed using a Nikon D5200 camera. For analysis of *pTKN2::nYFP*, *pCLAU::nYFP*, and *pTCSv2:3XVENUS* expression, dissected whole-leaf primordia were placed into drops of water on glass microscope slides and covered with cover slips. The pattern of YFP or VENUS expression was observed using a confocal laser scanning microscope (CLSMmodel SP8; Leica), with a solid-state laser set at 514 nm for excitation/ 530 nm for emission. Chlorophyll expression was detected at 488 nm excitation/ 700 nm for emission.

### GA content analysis

Giberellins were isolated and purified according to the method described by [80].

### Anthocyanin measurement

For anthocyanin measurement, plants were sprayed with the indicated CK concentrations three times a week for 3 weeks prior to analysis, starting upon emergence of the first leaf. Anthocyanins were extracted from the terminal leaflet of the third leaf by incubation overnight in methanol supplemented with a final concentration of 1% HCl. OD was measured in a plate spectrophotometer and anthocyanin content was calculated according to the following

formula: (OD530(0.25*OD660)), normalized to the starting tissue weight. Three technical replicates of 5 8 biological repeats were performed for each sample.

## Electrophoresis mobility shift assay (EMSA)

DNA probes were generated by end labeling of a 60-base single-stranded oligonucleotide using the DNA 3′ End Biotinylation Kit (Pierce 89818) and hybridization to complementary synthetic oligonucleotides (S1 Table) spanning binding sites for LA (GGNCC) which were identified using Sequencer 4.9, and generated with mutations disrupting the binding sites in the case of the mutant probe. Probes were generated by hybridizing the two complementary oligos by boil/cool. EMSAs were performed using the Light-Shift chemiluminescent EMSA kit (Pierce 20148). Briefly, 10 μL of purified recombinant MBP-LA fusion protein was incubated at room temperature in 1× binding buffer, 50 ng/μL poly(dI/dC), 2.5% glycerol, 0.05% Nonidet P-40, 50 fmol biotin-labeled probe, and 3.75 μg BSA for 30 to 40 min. The samples were resolved on 6% non-denaturing polyacrylamide gels, electrotransferred onto 0.45 μm Biodyne B nylon membrane (Pierce 7701), and cross-linked to the membrane. The migration of the biotin-labeled probe was detected on x-ray film (5-h exposure) using streptavidin–horseradish peroxidase conjugates and chemiluminescent substrate according to the manufacturer's protocol.

## Supporting information

**S1 Fig. (Supplement to Figure 1):** *CLAU* **and** *LA* **function in parallel pathways- leaflet quantification. (A-F)** Quantification of leaf complexity in genotypes with altered *CLAU* and *LA* expression levels. Graphs represent mean ± SE of six independent biological repeats. (**A-B**) Stacked bars of total leaflet number split into leaflet types, bars = SE. Statistical significance of differences in the total leaflet number was examined in a one-way ANOVA, p<0.0001. Different letters indicate significant differences between samples in an unpaired two-tailed t-test with Welch's correction. (**C-F**) Graphs representing each leaflet type separately. Statistical significance of differences was examined in a one-way ANOVA, p<0.0001. Different letters indicate significant differences between samples in a Tukey post hoc test (C) or a two-tailed t-test (D-F), (C) p<0.0443, (D) p<0.0039, (E) p<0. 045, (F) p<0.022.
(PDF)

**S2 Fig.** *CLAU* **expression in successive developmental stages of different leaves.** Expression level of *CLAU* was determined using RT-qPCR. **(A)** *CLAU* expression in the fifth plastochron in leaves 1 through 4. Leaves 1 and 2, which have reduced complexity, also have reduced CLAU expression. Graph represents mean ± SE of three independent biological repeats. Letters indicate significant differences in a one-way ANOVA with a Tukey post-hoc test, p≤0.037. **(B)** *CLAU* expression in successive leaf developmental stages, comparing the first and fifth leaves. Dashed line indicates expression trend. Graph represents mean ± SE of three independent biological repeats. Asterisks indicate significant differences in an unpaired, two-tailed t-test, p≤0.0019.
(PDF)

**S3 Fig. LA and CLAU expression domains and activity and TKN2. (A-D)** Phenotypes of leaves of the indicated genotypes. All leaves depicted are fully expanded fifth leaves. Bars = 2 cm.
(PDF)

**S4 Fig. (Supplement to Figure 3). TKN2 mediates the increased morphogenesis in** *CLAU*- **and** *LA*-**deficient backgrounds—leaflet quantification. (A-E)** Quantification of leaf complexity upon overexpression of TKN2-SRDX in the background of *CLAU* and *LA* deficiency. Graphs represent mean ± SE of at least three independent biological repeats. (**A**) Stacked bars

of total leaflet number split into leaflet types, bars = SE. Asterisks indicate significant differences of the total leaflet number from the background genotype (without TKN2-SRDX overexpression) in an unpaired two-tailed t-test with Welch's correction, p≤0.0335. (**B-E**) Graphs representing each leaflet type separately. Statistical significance of differences was examined in a one-way ANOVA, p<0.0001 (C-E). Different letters indicate significant differences between samples in a two-tailed t-test (B,C,E) or a Tukey post hoc test (D), (B) p = ns (no significant differences), (C) p<0.0249, (D) p<0.0052, (E) p<0.049.
(PDF)

**S5 Fig.** TKN2 is expressed at the leaf margin in CLAU and LA deficient backgrounds (A-D) Expression pattern of the TKN2 promoter fused to YFP in WT M82 plants. (A) Vegetative meristem (m) and two youngest leaf primordia (P1, P2). (B) Floral meristem (FM), sympodial inflorescence meristem (SIM). (C-D) Weak expression is observed in P3/P4 leaf primordia (area of the leaf primordium depicted in C is indicated in the inset). (E) Confocal micrographs of pTKN::nYFP in the fourth plastochron (P4) in indicated genotypes. TKN2 promoter activation increases in the leaf margin in CLAU and LA deficient backgrounds. The pattern of YFP expression was detected by a confocal laser scanning microscope (CLSM model SP8; Leica), with the solid-state laser set at 514 nm for excitation and 530 nm for emission. Chlorophyll expression was detected at 488nm excitation/ 700nm emission. Bars = 200 um.
(PDF)

**S6 Fig. (Supplement to Figure 5)/** *clausa* **has reduced sensitivity to GA treatment–leaflet quantification.** (**A-E**) Quantification of leaf complexity following GA treatment in WT and *clausa*. Graphs represent mean ± SE of at least three independent biological repeats. (**A**) stacked bars of total leaflet number split into leaflet types, bars = SE. Statistical significance of differences in the total leaflet number was examined in a one-way ANOVA, p<0.0001. Different letters indicate significant differences between samples in an unpaired two-tailed t-test with Welch's correction (p<0.015). (**B-E**) Graphs representing each leaflet type separately. Statistical significance of differences was examined in a one-way ANOVA, p<0.0001. Different letters indicate significant differences between samples in a tukey post hoc test (B,E) or a two-tailed t-test (C,D), (B) p<0.0043, (C) p<0.011, (D) p<0.02, (E) p<0.03.
(PDF)

**S7 Fig. (Supplement to Figure 5).** *clausa* **has reduced sensitivity to GA treatment–leaflet quantification.** (**A-E**) Quantification of leaf complexity following GA treatment in WT and *clausa*. Graphs represent mean ± SE of at least three independent biological repeats. (**A**) stacked bars of total leaflet number split into leaflet types, bars = SE. (**B-E**) Graphs representing each leaflet type separately. Statistical significance of differences was examined in a one-way ANOVA, p<0.0001 (C-E). Different letters indicate significant differences between samples in a two-tailed t-test, (B) p = ns (no significant differences), (C) p<0.0255, (D) p<0.0398, (E) p<0.0212.
(PDF)

**S8 Fig. CK response is reduced at the leaf margin upon** *LA* **overexpression.** (A-F) Confocal micrographs of *TCSv2::3XVENUS* in successive developmental stages in indicated genotypes. *TCSv2* driven signals are reduced in *La-2/+* in the fifth plastochron. The pattern of VENUS expression was detected by a confocal laser scanning microscope (CLSM model SP8; Leica), with the solid-state laser set at 514 nm for excitation and 530 nm for emission. Chlorophyll emission was detected at 488nm excitation/ 700nm emission. Bars = 100 um. (G-H) Quantification of *TCSv2* driven Venus fluorescence in indicated developmental stages of indicated genotypes. (J) Mean fluorescence % of WT meristem signal; (K) Corrected total fluorescence

(CTF) was calculated as [integrated density–(area*background mean fluorescence)]- presented as % of WT meristem CTF. Quantifications were done on at least 5 plants from each genotype in 3 experiments. Asterisks indicate significance from WT of the same developmental stage in a two-tailed t-test, $*p<0.05$, $**p<0.01$, $***p<0.001$, ns = not significant.
(PDF)

**S9 Fig. *LA* deficiency enhances CK sensitivity to anthocayanin accumulation.** Anthocyanin content in WT and altered *LA* genotypes with or without CK treatment was determined by measuring optical density following methanolic extraction. Graph represents mean ± SE of five independent biological repeats. Letters indicate significant differences in an unpaired, two-tailed t-test with Welch's correction, $p \leq 0.038$.
(PDF)

**S10 Fig. LA binds *in vitro* to the promoters of TRRs. (A)** LA binds *in vitro* to the promoters of TRRs. Recombinant MBP-LA caused a shift in the PAGE migration of biotinylated promoter derived probes (Probe-B) of the indicated TRRs. **(B)** Specific binding of *LA* to the TRR3/4 promoter. Recombinant MBP-LA caused a shift (**s**) in the PAGE migration of the biotinylated promoter derived probe (**p**) of TRR3/4. A non-biotinylated probe was able to compete with the biotinylated probe in a quantity depended manner (**c1-c3**). MBP (**MBP empty**) was used as a control protein and does not cause a probe shift in PAGE. **(C)** A biotinylated probe with a mutation in the binding site (**m**) does not shift in the presence of MBP-LA, and cannot compete with the WT probe (**s+m1-m3**).
(PDF)

**S11 Fig. *CLAU* and *LA* function in parallel pathways in additional developmental processes. (A-H)** Genetic interactions between genotypes with altered *CLAU* and *LA* expression levels and GA levels or response. *La-2/+*: a semi-dominant LA allele with increased and precocious expression due to miR319 resistance. *FIL>>CLAU*: *CLAU* upregulation. Increased GA levels (GA application) or response: in *procera (*pro) loss-of-function. Bars = 2 cm. Red and white arrowheads represent flowers and leaf-like structures, respectively. **(I)** Quantification of inflorescence complexity in the indicated genotypes and treatments. Graphs represent mean ± SE of at least 4 independent biological repeats. Letters indicate significant differences between samples in a one-way ANOVA with a Tukey post-hoc test, $p<0.0009$.
(PDF)

**S1 Table. Primers used in this work.**
(PDF)

**S1 Data. Lists of upregulated DEGs in all genotypes- excel file.**
(XLSX)

**S2 Data. Differential KEGG pathways in all genotypes- excel file.**
(XLSX)

**S3 Data. Morphogenesis and differentiation genes from Ichihashi et al., 2014- excel file. Classical morphogenesis genes identified in the data are highlighted.**
(XLSX)

**S4 Data. Representation factor and analysis of the overlap of "morphogenesis" and "differentiation" genes from Ichihashi et al., 2014, with CLAU and LA genotypes.**
(PDF)

## Acknowledgments

We thank members of the Ori, Efroni, and Bar groups for their support.

## Author Contributions

**Conceptualization:** Alon Israeli, Yogev Burko, Naomi Ori, Maya Bar.

**Funding acquisition:** Alon Israeli, Naomi Ori.

**Investigation:** Alon Israeli, Yogev Burko, Sharona Shleizer-Burko, Iris Daphne Zelnik, Noa Sela, Mohammad R. Hajirezaei, Alisdair R. Fernie, Takayuki Tohge, Naomi Ori, Maya Bar.

**Supervision:** Naomi Ori, Maya Bar.

**Writing – original draft:** Alon Israeli, Yogev Burko, Naomi Ori, Maya Bar.

**Writing – review & editing:** Alon Israeli, Yogev Burko, Naomi Ori, Maya Bar.

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
