## [Decision Letter · Decision Letter 0]

9 Feb 2021

Dear Dr Israeli,

Thank you very much for submitting your Research Article entitled 'Coordinating the morphogenesis-differentiation balance by tweaking the cytokinin- gibberellin equilibrium' to PLOS Genetics.

The manuscript was fully evaluated at the editorial level and by independent peer reviewers. The reviewers appreciated the attention to an important problem, but raised some substantial concerns about the current manuscript. Based on the reviews, we will not be able to accept this version of the manuscript, but we would be willing to review a much-revised version. We cannot, of course, promise publication at that time.

If you decide to revise the manuscript for further consideration at PLOS Genetics, please aim to resubmit within the next 60 days, unless it will take extra time to address the concerns of the reviewers, in which case we would appreciate an expected resubmission date by email to plosgenetics@plos.org.

[LINK]

We are sorry that we cannot be more positive about your manuscript at this stage. Please do not hesitate to contact us if you have any concerns or questions.

Yours sincerely,

Sarah Hake

Associate Editor

PLOS Genetics

Claudia Köhler

Section Editor: Plant Genetics

PLOS Genetics

Editor's comments:

One reviewer was enthusiastic and not very critical, the other reviewer was much more critical, but the criticism can probably be addressed in a resubmission. For example, the temporal action is mentioned but not demonstrated. Can you provide more information, (perhaps a cartoon) of the timing of expression for the different promoters?

This reviewer also is concerned about the fact you treat all leaflets the same and just count them. Is it possible to address that criticism?

Other criticisms should also be considered.

As the editor and one who doesn’t read every tomato leaf paper, I enjoyed it and found most of the figures stunning. I was not as pleased with the transcriptome data as it is very hard to compare across different experiments. Ideally, the different genotypes would be grown at the same time and carefully harvested to make sure similar sampling was carried out. I gathered you took already published data. Comparing microarray data with RNAseq data is not great. And we have no idea what tissue was used, what stage, etc.

Here are some additional points you need to address.

What is the BLS promoter? As mentioned already, the reader’s need reminding of the timing of different promoters.

In Figure 5, there is quantification for adding GA, but not for the other panels. Please add.

Figure S2B, D do not seem to correspond to the statement “This is in accordance with CLAU being expressed later than LA during the morphogenetic window, and with the effect of stage-specific expression of TKN2 (Figure S2B, D and (20,29,30,39,44)).”

Please say what SlGA20ox-1 is.

Please tell us about TCSv2 which shows up in Figure 6.

I don’t see the data for this statement “morphogenesis genes were significantly enriched in

morphogenetic genotypes clausa, la-6, and overexpression of miR319 or TKN2, while

differentiation genes were significantly depleted in these genotypes” in figure 7. Does it require looking at the supplemental data? Looking at supplemental data sets means that no one sees the data. If there are important comparisons, they need to be more accessible.

Reviewer's Responses to Questions

**Comments to the Authors:**

Reviewer #1: Tomato leaf, with the formation of multiple leaflets, is one of the major model of compound leaf morphogenesis and is used to understand how complex shapes form in plants. The authors addressed here the question of the relationship between different pathways mediated by transcription factors, LA, CLAU and TKN2 and hormones, mostly cytokinins and gibberellins during tomato leaf morphogenesis. There is already an extensive literature about this type of questions, and the general principle of the effects of these pathways (ie that they affect the balance between morphogenesis and differentiation and hence the possibility to elaborate complex shapes).

To analyse the relationship between LA, CLAU and TKN2, the authors analyse an impressive number of lines in which the activity of these gene is modified using different strategies (loss-of-function mutant, gain-of-function, ectopic expression from heterologous promoters, expression of repressor fusion proteins). The phenotypic characterisation of these lines mostly remains at the mature stage level and some of the conclusions are contradictory (see below). Then, the authors show that CLAU modifies the GA content and response. A reduced sensitivity of LA to CK is shown, extend observations previously made in Arabidopsis. Finally, trancrriptomics provides data about the overlap between LA, CLAU and TKN2 pathways and their effect on the expression of cohorts of genes involved in morphogenesis and differentiation.

The paper suffers from some major weaknesses (see below). It provides a refining of the interaction between different pathways, but as such this is not a major contribution to the field. Instead, some data such as the different temporal action of these different pathways are mentioned in the paper, but are not supported enough by the data.

Major comments :

Tomato develop different types of leaflets (primary, secondary, interacallary) that form at different moments on the leaf primordium and at different relative positions. Such distinction between the different types of leaflets in not made here (eg Fig1MN) Also, the early stages of the leaf morphogenesis, that would give some hints about the dynamic process of leaf shape elaboration are not shown. It is therefore difficult to understand the developmental defects that lead to the perturbed mature leaf morphologies that are shown. Typically, la-2+/ clausa and FIL>>CLAU or clausa and FIL>MIR319 are described as having the same leaflet number (Fig 1MN), yet their morphologies are different. With the data provided, it is unclear if these lines indeed develop in a similar way or if they reach a similar leaflet number by two totally distinct trajectories. That going to the cellular level is important was shown by the same group in Shani et al., 2009, where they showed that expressing TKN2 or TKN2-SRDX under the FIL promoter leads to similar simple leaves which are however due to delayed or precocious maturation of the primordium

The interpretation of the role of TKN2 is not correct. For instance, it is mentioned in the abstract that “Maintaining a prolonged morphogenetic phase in early stages of compound-leaf development in tomato is dependent on delayed activity of several factors that promote differentiation, including … the morphogenesis promoting activity of the transcription factor TKN2 ». This conclusion is based on the effects of its overexpression, which does not necessarly mean that this is the role of the endogenous TKN2 gene.

The way observations following TKN2-SRDX expression are interpreted must also be more cautious.

First, because, TKN2-SRDX overexpression does not fully mimic a TKN2 mutant. In other publications of the groups it is stated that TKN2-SRDX may down-regulate expression of TKN1 to 4.

Second, because if promoters other than its own promoter are used, its expression pattern/level can be modified and the TKN2-SRDX fusion may repress targets that may not be the true TKN2 targets. Therefore, stating as in page 12 that “In addition, reducing TKN2 targets from the LA expression domain (LA>>TKN2-SRDX) resulted in similar phenotypes to those of La-2/+ mutants (Figure S2B), which stresses the important role of TKN2 downstream of LA. » is an overinterpretation.

Page 1, it is concluded that CLAU and LA act through parallel pathways. « These genetic analyses demonstrate that the effect of CLAU and LA on leaf development is partially additive, indicating that they promote differentiation in parallel pathways ». But later, it is concluded that LA defines the expression window of CLAU, which rather indicates an overlap between both pathways. And the authors also conclude that LA and CLAU converge on CK/GA. Therefore, the relationship between both pathways is not clear. One reason for this could be that for the genetic analysis (Fig1), only partial loss-of-functions are used for LA and no real knock-out.

Minor comments.

Page 11. « To better understand the genetic regulation of the balance between morphogenesis and differentiation, we examined the interaction between the TFs CLAU and LA » is very misleading as it suggests interaction between these proteins. Here the genetic interactions are analysed.

Page 12. The title “Common TKN2-promoted pathways mediate extended morphogenesis » is not clear.

Page 12. Remind the reader of the expression pattern of the different promoters used, eg BLS.

Page15. Indicate the tissue that was sampled for the RNAseq experiments.

Page 16. “DEGs confirm dependencies between the LA genotypes, with between about a third to half of the genes significantly upregulated in La-2/+ being significantly downregulated in la-6 and upon miR319 overexpression (Figure 7, Supplementary Data 1). » This is not shown in figure 7.

Some of the references used for TKN2 are not appropriate. The mentionned papers do not included this gene. Eg ref 40, 41, 43. In fact this gene has only poorly been characterised.

Page 20 : « Our results demonstrate a "gradient" of transcription factor activity…that is translated to leaflet number ». Which gradient of transcription factors are the authors talking about?

Reviewer #2: The manuscript by Israeli et al. entitled “Coordinating the morphogenesis-differentiation balance by tweaking the cytokinin-gibberellin equilibrium” investigates the relationship between the several key transcription factors LA, CLAU and TKN2, and the plant hormones GA and CK, in the regulation of the morphogenesis-differentiation balance. Genetic analyses revealed that the expression levels of LA in different genetic backgrounds determined the expression domain and level of CLAU to affect the final leaf shape of tomato. The function of LA and CLAU is relied on the activity of target TKN2. The clausa mutant has an altered GA profile and reduced sensitivity to GA treatment, but LA reduces leaf margin sensitivity to CK. They also revealed that plant transcription factors and hormones GA and CK are coupled into tuning the final compound leaf shape.

The text is concisely written, and data are clear and solid. I have a few points to be addressed in order to improve the quality of this work.

（1） In the Figure 1J and 1M, they are probably from different stages of compound leaves. Please indicate the exact stages of each photo. For example, In the Fig 1J, the leaflet number of clausa mutant is less than that of Fig M.

（2） In the Figure 7 legend ” Global expression analyses of LA, CLAU and TKN2 ” should being changed to “Global expression analyses of LA, CLAU and TKN2”.

**Have all data underlying the figures and results presented in the manuscript been provided?**

Reviewer #1: Yes

Reviewer #2: Yes

PLOS authors have the option to publish the peer review history of their article (what does this mean?). If published, this will include your full peer review and any attached files.

Reviewer #1: No

Reviewer #2: **Yes: **Jianghua Chen

---

## [Editor Report · Decision Letter 1]

29 Mar 2021

Dear Dr Israeli,

Thank you very much for submitting your Research Article entitled 'Coordinating the morphogenesis-differentiation balance by tweaking the cytokinin-gibberellin equilibrium' to PLOS Genetics.

The manuscript was fully evaluated at the editorial level and by independent peer reviewers. The reviewers appreciated the attention to an important topic but identified some concerns that we ask you address in a revised manuscript

We therefore ask you to modify the manuscript according to the review recommendations. Your revisions should address the specific points made by each reviewer.

[LINK]

Yours sincerely,

Sarah Hake

Associate Editor

PLOS Genetics

Claudia Köhler

Section Editor: Plant Genetics

PLOS Genetics

The authors have done a great job addressing concerns of reviewers and myself. There are a couple places where the figure is not correctly mentioned in the text, so please go through and check that carefully. For example, Figure 6C is mentioned incorrectly and there were perhaps a couple more.

The authors mention the reduction of TcS in La-2/+. Because there are no leaflets, there is definitely less of this reporter, but is there significantly less of the reporter in the margin that does exist? Do you have a way to quantify the reduction?

You bring up anthocyanin on page 28. That seems totally out of the blue and I am not sure it is needed. (Figure S9).

---

## [Editor Report · Decision Letter 2]

6 Apr 2021

Dear Dr Israeli,

We are pleased to inform you that your manuscript entitled "Coordinating the morphogenesis-differentiation balance by tweaking the cytokinin-gibberellin equilibrium" has been editorially accepted for publication in PLOS Genetics. Congratulations!

Yours sincerely,

Sarah Hake

Associate Editor

PLOS Genetics

Claudia Köhler

Section Editor: Plant Genetics

PLOS Genetics

Comments from the reviewers (if applicable):

**Data Deposition**

http://datadryad.org/submit?journalID=pgenetics&manu=PGENETICS-D-20-01955R2

**Press Queries**

---

## [Editor Report · Acceptance letter]

16 Apr 2021

PGENETICS-D-20-01955R2 

Coordinating the morphogenesis-differentiation balance by tweaking the cytokinin-gibberellin equilibrium 

Dear Dr Israeli, 

We are pleased to inform you that your manuscript entitled "Coordinating the morphogenesis-differentiation balance by tweaking the cytokinin-gibberellin equilibrium" has been formally accepted for publication in PLOS Genetics! Your manuscript is now with our production department and you will be notified of the publication date in due course.

With kind regards,

Katalin Szabo

PLOS Genetics

On behalf of:
